# Sequence Variant Analysis of the *APOCII* Locus among an Arab Cohort

**DOI:** 10.3390/ijms242216293

**Published:** 2023-11-14

**Authors:** Suzanne A. Al-Bustan, Maryam H. Alrashid, Ahmad E. Al-Serri, Babitha G. Annice, Hussain M. Bahbahani

**Affiliations:** 1Department of Biological Sciences, College of Science, Kuwait University, Farwaniya 85700, Kuwait; maryam.alrashid@ku.edu.kw (M.H.A.); babitha.george@ku.edu.kw (B.G.A.); hussain.bahbahani@ku.edu.kw (H.M.B.); 2Unit of Human Genetics, Department of Pathology, Faculty of Medicine, Kuwait University, Hawally 46300, Kuwait; ahmad.alserri@ku.edu.kw

**Keywords:** *APOCII*, sequencing, genetic variants, T2DM, hypertriglyceridemia, Arabs, Kuwait

## Abstract

Apolipoprotein CII (ApocII) plays a key role in regulating lipoprotein lipase (LPL) in lipid metabolism and transport. Numerous polymorphisms within *APOCII* are reportedly associated with type 2 diabetes mellitus (T2DM), dyslipidemia, and aberrant plasma lipid levels. Few studies have investigated sequence variants at *APOCII* loci and their association with metabolic disorders. This study aimed to identify and characterize genetic variants by sequencing the full *APOCII* locus and its flanking sequences in a sample of the Kuwaiti Arab population, including patients with T2DM, hypertriglyceridemia, non-Arab patients with T2DM, and healthy Arab controls. A total of 52 variants were identified in the noncoding sequences: 45 single nucleotide polymorphisms, wherein five were novel, and seven insertion deletions. The minor allele frequency (MAF) of the 47 previously reported variants was similar to the global MAF and to that reported in major populations. Sequence variant analysis predicted a conserved role for *APOCII* with a potential role for rs5120 in T2DM and rs7133873 as an informative ethnicity marker. This study adds to the ongoing research that attempts to identify ethnicity-specific variants in the apolipoprotein gene loci and associated *LPL* genes to elucidate the molecular mechanisms of metabolic disorders.

## 1. Introduction

Apolipoproteins (lipid-free) are lipid-binding proteins that circulate in the animal bloodstream. They are involved in the transport of triglycerides, phospholipids, cholesterol, and cholesteryl esters between organs. Apolipoproteins combine with lipids to form several classes of lipoproteins that vary in their density including chylomicrons, very low-density lipoproteins (VLDLs), high-density lipoproteins (HDLs), and low-density lipoproteins (LDLs). Their primary function is to signal and target lipoproteins to specific tissues or to activate enzymes that act on lipoproteins. There are two major gene clusters for apolipoproteins located on chromosome 19, consisting of *APOE*, *APOCI*, and *APOCII* and chromosome 11, consisting of *APOAI*, *APOAIV*, *APOCIII*, *APOAIV*, and *APOAV* [1,2]. The *APOE*/*CI*/*CII*/*CIV* gene cluster is located on chromosome 19q13.2 and spans approximately 48 kbp [3]. This gene cluster has numerous polymorphisms and is associated with type 2 diabetes mellitus (T2DM), dyslipidemia, and other complex diseases, with varying degrees of association with different plasma lipid levels.

ApocII is mainly synthesized in the intestine and liver for incorporation into chylomicrons, VLDLs, and HDLs. The preprotein contains 101 amino acids including a 22 amino acid signal peptide that is cleaved, resulting in a mature protein consisting of 79 amino acids [4]. This protein is an important activator of lipoprotein lipase (LPL) and of the lipolytic metabolism of triglycerides (TGs) [5]. However, its overexpression has been suggested to have the opposite effect, leading to increased TG levels [6,7]. *APOCII* itself is only 3.58 kb, starting at nucleotide position 45,449,239 and ending at 45,452,822 [6]. This gene comprises four exons and three introns. Numerous single nucleotide polymorphisms (SNPs) have been identified and documented in GenBank and Ensembl. Few studies have investigated sequence variants at the *APOCII* loci and their association with metabolic disorders, including T2DM [7,8,9,10,11,12].

The general lifestyle of many Kuwaitis includes a high-fat diet and minimal physical activity, thereby increasing their risk of developing metabolic syndrome and subsequent heart disease. The reported prevalence of subjects that are overweight, obese, and have metabolic syndrome is 80.4%, 47.5%, and 36.2%, respectively, in the Kuwaiti adult population [13]. Moreover, there were 407,530 Kuwaiti cases with diabetes in 2013, with a prevalence rate of 18.3% [14] that had slightly increased from 1995 [15,16]. These unusually high percentages may reflect defects in mechanisms underlying lipid transport and metabolism, resulting in abnormal plasma lipid levels that may involve *APOCII* [5,17].

In-depth analysis using deep sequencing or similar techniques [18] allows the identification of different alleles (common, rare, and very rare) that may significantly contribute to the complex nature of metabolic disorders [19]. Moreover, many genetic variants associated with diseases have been observed in noncoding regions, particularly introns [19,20]. This may indicate that such variants affect gene regulation, pre-mRNA splicing, and mRNA stability rather than protein function [21,22]. Furthermore, noncoding variants at related gene loci may interact with the molecular etiology of the disease. Identifying such variants is important to understand how several common alleles influence the variation. Each allele may have a modest effect, and when combined, may exert an influence on the manifestation of T2DM.

Resequencing studies and mutation analysis in other populations and diseases have reported unconfirmed associations between different SNPs [5,7,8,10,17,23]. However, the available data are limited.

This study aimed to identify and characterize genetic variants by sequencing the full *APOCII* locus and its flanking sequences in a sample of the Kuwaiti Arab population to continue previous efforts to characterize gene variants at apolipoprotein gene loci [24,25,26]. The study also aimed to establish a variant database and possibly a diagnostic panel for future studies related to metabolic disorders, including dyslipidemia and T2DM, which may be ethnicity-specific.

## 2. Results

Full gene and flanking sequences (4980 bp) were successfully generated for the 200 Kuwaiti Arab samples by Sanger sequencing using custom-designed primer sets (*n* = 10) that amplified overlapping regions averaging between 385 and 413 bp in length. The fully annotated Kuwaiti Arab *APOCII* sequence, including upstream and downstream flanking sequences (chromosome 19: 44945371–44950166), was deposited in GENBANK. The accession number for the deposited sequence with the identified SNP variants is *APOCII* MT073872 (BankIt 2306226).

### 2.1. APOCII Sequence Variant Identification and Analysis

Sequence analysis and alignment with the reference sequence (NG_008837.1) identified 52 sequence variants along the target region and among all the samples analyzed (*n* = 200). Five novel sequences, single nucleotide variant (SNV), were identified from the 52 variants (Table 1). None of the identified variants had an apparent functional role and were classified as simple sequence variants based on the Ensembl genome browser Build GRCh38 (https://grch37.ensembl.org/index.html accessed on 1 April 2020). There are 5426 reported variants for the region sequenced in Ensembl, wherein 4359 and 574 are SNVs and insertion deletions (INDELs), respectively. The remaining variants are categorized as deletions (99), insertions (59), somatic single-nucleotide variants (325), somatic insertions (5), and sequence alterations (5).

The highest number of variants were observed in intron 1 (*n* = 30), followed by the upstream region including *APOCIV* 3′UTR and *APOCII* 5′UTR (*n* = 8), and the least number of identified variants were in *APOCII* 3′UTR (*n* = 3). Most of the identified variants were SNPs (*n* = 45), whereas seven previously reported INDELs were identified (Table 2). There were five deletions and two insertions. No variants were identified in the coding region (Exons 1–4) of the *APOCII* locus.

The 45 identified variants included transitions and transversions in noncoding regions, with nearly twice as many transitions (*n* = 31) as transversions (*n* = 14) observed at the *APOCII* locus. The highest frequency of variants was observed in intron 3 (Figure 1) and no sequence variants were identified in intron 2.

Most variants were either “common” (minor allele frequency (MAF) ≥ 0.05) or “very rare” (MAF ≤ 0.01). The MAF distribution was compared to the reported global MAF and that of four major populations (Figure 2), excluding the five novel variants. Notably, most variants were either “very common” (MAF ≥ 0.01) or “very rare” (MAF ≤ 0.01). The five novel SNVs were identified in single samples as heterozygous in different groups (Table 3). Two SNVs, KUA ApoCII1 (5′UTR g.678 G>A) and KUA ApoCII2 (intron 1 g.1776 T>C), were identified in a non-Arab and an Arab patient with diabetes, respectively. 

### 2.2. Genotype Analysis among the Studied Groups

The genotype and allele frequencies (Appendix A) for 42 variants were in equilibrium (Hardy–Weinberg equilibrium (HWE) *p* ≥ 0.05). The 10 remaining variants had genotype frequencies that deviated from HWE in the different groups. The genotypes of two variants, rs2288912 and rs5120, that deviated from HWE were in the non-Arab T2DM group (*p* = 0.003, *p* = 0.041), whereas rs9304644 was the only genotype that deviated from HWE in the Arab T2DM group (*p* = 0.23). In the three Arab groups, the genotypes of four variants deviated from HWE (*p* < 0.05), including rs10421404, rs7257468, rs7258345, and rs7257476. Three variants, rs3745152, rs1130742, and rs4803776, had genotypes that significantly (*p* < 0.05) deviated from HWE in all studied groups, owing to the very low numbers of heterozygotes. Genotype distributions of the four groups were compared and analyzed using a contingency table. There were no significant (*p* ≥ 0.05) differences in the genotype distribution among the four studied groups.

### 2.3. Differences in the MAF Distribution among the Studied Groups

Differences in the MAF of variants were estimated between the four groups to identify variants that had different frequency distributions between the sequenced subgroups. The SNVs that had an MAF ≤ 0.05 in more than two groups were excluded. Eighteen SNPs were analyzed for differences in MAF among the four groups (Figure 3). There was a difference in the MAF for rs12709889 among all four compared groups, wherein the non-Arab T2DM group had the highest frequency (MAF = 0.22), whereas the control group had an MAF of 0.09.

### 2.4. Haplotype Construction and Analysis

Haploview 4.2 was used for linkage disequilibrium (LD) analysis of the *APOCII* genetic variants in 200 Kuwaiti samples. Sequence variants with MAF < 0.05 were excluded from the LD analysis to avoid statistical error and false results. Therefore, only 15 SNPs with two INDELs, rs150448996 Ins -/T and rs10622462 Ins -/CCT (Figure 4), were selected. Two haplotype blocks were created based on the confidence interval logarithm from 15 SNPs and two INDELs. Contrary to the first haplotype block, the second haplotype was constructed from SNPs that deviated from the HWE at *p* < 0.001, with a borderline r^2^ of 0.76. The strongest LD was found between rs9304644 C>T and rs7256684 A>G, with r^2^ = 0.95 (Table 4). One INDEL, rs10622462, strongly correlated with rs7256684, rs9304644, and rs9304646, with an r^2^ of 0.93, 0.90, and 0.86, respectively.

Linkage disequilibrium and haplotype associations were analyzed and compared for three case groups: non-Arab T2DM (*n* = 50), Arab T2DM (*n* = 50), hypertriglyceridemia (HTG, *n* = 50), and Arab controls (*n* = 50), according to standard criteria. The LD plot created from the HTG and Arab control groups was similar to that from the total population in Figure 4, whereas a smaller haplotype block with a reduction in representative SNPs was created in the non-Arab T2DM versus the Arab control groups (Figure 5). In all cases, the strongest LD remained between rs9304644 and rs7256684, except for the HTG data, where the strongest LD was found with two SNP pairs at an r^2^ of 1. The first SNP pair (rs9304644 and rs4803774) had an r^2^ of 0.81 in the total population (Table 5). The second pair, rs10419086 and rs10420434, located at a distance of 338 bp, had r^2^ of 0.71 in the total population.

Multiple analyses of haplotype association revealed a significant negative association with the non-Arab T2DM group. Two haplotypes in the first block had a significant higher frequency in the Arab control group at *p* = 0.0125 for the “CCAGCGGA” haplotype and *p* = 0.0145 for the “CGAGCGGA” haplotype (Table 5). The haplotype blocks include rs9304646, rs11879392, rs10419086, rs4803774, rs4803775, rs10420434, rs7256684, and rs5120 (Figure 5A). Haplotype associations for SNPs that deviated from HWE in the second block (rs3745152 and rs4803776) showed no significant association with Arab T2DM or HTG. 

### 2.5. Principal Component Analysis

Principal component analysis did not show clear genetic distinction among the four analyzed groups along the first three principal components, which accounted for 2.6%, 1%, and 0.08% of the total variation (Figure 6).

## 3. Discussion

This study reports the identification and characterization of the full *APOCII* locus and its flanking sequences in a sample of the Kuwaiti Arab population, including healthy controls, patients with T2DM, patients with HTG, and non-Arab patients with T2DM. A total of 52 variants were identified only in the noncoding sequences: 45 variants, of which five were novel, and seven INDELs. The 52 identified variants did not have a functional role as predicted by the in silico analysis documented in the Ensembl database. This suggested that the gene locus was protected under natural selection. The variants identified in our study were mostly localized in intron 1, with the largest region spanning 19:44946075–19:44948478 (2403 kbp). No variants were found in the coding regions of the 200 sequenced samples. Sequence analysis of the *APOCII* locus is scarce and was previously conducted in the Chinese [27], non-Hispanic white, African black [10], and Turkish [12] populations. A review of studies reporting coding variants in patients with rare apolipoprotein C2 deficiency also suggested the conserved nature of this gene locus [7].

Rare loss-of-function variants in *APOCII* can lead to decreased interactions between LPL and TG-rich lipoproteins [28]. A limited number of studies [7,9,10,27] have reported an association between coding/noncoding sequence variants at the *APOCII* locus and metabolic disorders, including T2DM, which supports our findings. Our sequence analysis did not show any genetic variants in intron 2, unlike a study conducted in Turkish patients with HTG, which reported six variants in intron 2 [12]. It is possible that variants in intron 2 are also conserved; those identified in a Turkish study were found in a few patients with severe clinical HTG [12].

Furthermore, a study that used tagged SNPs to investigate the association of variants in the *APOE/C1/C4/C2* gene cluster with plasma lipid concentration reported a reduced risk of coronary heart disease in UK men harboring rs5127 [8]. Notably, this variant was potentially associated with T2DM in non-Arabs, but this was not confirmed and requires further investigation. A confirmed association between noncoding *APOCII* variants and metabolic disorders has not been reported, which is consistent with our findings. This again suggests that the gene locus is highly conserved.

The MAF of the 47 previously reported variants were similar to that reported for the global MAF and in major populations. The lowest reported number of total variants was observed in the American and South and East Asian populations. Out of the total variants, 27 had an MAF ≥ 0.05 and were similar to the other reported frequencies in the major populations. One reported variant (rs71338739) was observed in 50% of the sequenced Kuwaiti samples. This variant is a single deletion of the tetranucleotide repeat TGTG at nucleotide position 44946192 in intron 1 (g.955–959). This was not previously reported in the global population, or the four populations documented in GenBank; however different size deletions/insertions at the same locus were reported (https://www.ncbi.nlm.nih.gov/snp/rs71338739#frequency_tab; accessed 1 May 2023). This variant should be further investigated for its potential use in panels for population stratification and DNA forensics. Furthermore, in this study, five novel SNVs were identified that could be classified as private variants occurring in only the noncoding sequences (UTR and intron 1) of the *APOCII* locus. These SNVs were considered uninformative regarding their association with metabolic diseases or population genetic structures. Similar findings have been reported in non-Hispanic Whites (NHW) and African Blacks (AB). Pirim et al. [10], reported eight novel variants at the *APOCII* locus in NHW and 17 novel variants in AB; all variants were localized in intron 1, 3, and the 3′ flanking.

The MAF of the variants detected in *APOCII* can vary widely among different populations (Table 6). For instance, rs2288912 and rs9304644 showed similar MAF regardless of the population investigated. In contrast, rs12721076 exhibited an MAF ≥ 0.05 in the global population, admixed Americans (AMR), Kuwaitis (KWT), NHW, and South Asians (SAS), whereas in AB the MAF was ≤0.05. Another striking example is rs11879392, for which MAF varied from 0.027 in AB to 0.26 in SAS. 

The SNP rs1130742 in the *APOCII* 3′UTR had a much lower MAF (0.05) in the studied population than that in the global population (0.3285) and other reported major populations, with the highest being reported in the east Asian population (0.503). Two variants (rs7257095 and rs12709886) were “rare” and similar to those reported in global and major populations. The remaining 23 variants were all “very rare” with frequencies similar to those reported in the global and major populations. Furthermore, 10 variants deviated from HWE owing to low heterozygote counts. This could be explained by either the cultural practices of the Kuwaiti Arab population in encouraging consanguineous marriages to increase homozygotes for ancestral alleles, or the possibility that the variants accumulated in relation to the phenotype. For example, HWE for rs9304644 deviated only in Arab patients with T2DM, whereas HWE for rs2288912 and rs5120 deviated only in non-Arab patients with T2DM. These variants were not previously reported to be associated with T2DM and should be further investigated in a powered cohort to determine whether the deviation is related to their association with the risk for T2DM.

This study found no statistical differences regarding the genotype distribution between the four groups (*p* ≥ 0.05). Other studies [30,31,32,33,34,35,36,37] also reported no significant differences between *APOCII* variants and T2DM and HTG (Appendix A). This also supports the suggestion that the gene locus may be under selective pressure owing to the vitality of its function in regulating lipid levels. In addition, principal component analysis showed that the identified *APOCII* variants did not cluster into groups independently (Figure 6). This indicates that these variants may not be directly involved in the molecular etiology of T2DM.

We also attempted to identify potential variants that could be further investigated for their association with T2DM and HTG using differences in MAF. We identified possible (although not significant) roles for rs12721060, rs1019828365, rs5120, and rs7258345 in the risk of developing HTG. Therefore, the rs5120 SNP warrants further investigation. A previous study conducted on NHW and AB reported that three variants (APOC2/rs5120, APOC2/rs10422888, and APOC2-C4/rs12709885) at the *APOCII* locus were significantly associated with LDL and require further investigation to support our provisional findings on rs5120 [10].

The haplotype analysis revealed several noteworthy findings. The strongest LD was observed for rs9304644 and rs7256684 in the T2DM group. In addition, a significant negative association was found between the haplotype of the non-Arab T2DM group and the other groups. The haplotype block included the previously implicated SNPs rs9304646, rs11879392, rs10419086, rs4803774, rs4803775, rs10420434, rs7256684, and rs5120. This suggests an accumulative effect of the variants, which warrants further investigation. However, Ken-Dror (2010) reported different LD values between different SNPs, suggesting that different LD and haplotypes are found in different populations and ethnic groups, as reported by Pirim et al. [10].

## 4. Materials and Methods

### 4.1. Sample Description

The samples used in this study were obtained from volunteers who visited general hospitals and polyclinics in Kuwait City Governate for routine check-up during 2014–2017. Blood samples (10 mL) were collected by a certified nurse into EDTA tubes via venipuncture for serum lipid level analysis and whole blood for genomic DNA (gDNA) extraction. *APOCII* sequence variant analysis was performed on 200 samples (94 males and 106 females) consisting of samples with Arab ancestry with a confirmed clinical diagnosis of T2DM (*n* = 50) or HTG (*n* = 50), non-Arab ancestry with confirmed T2DM (*n* = 50), and healthy controls of Arab ancestry (*n* = 50). (Table 7). Controls were matched and selected based on age, sex, and ethnicity. The inclusion criteria for the full cohort (*n* = 200) were documented metabolic disorders at the time of sample collection, known ancestry based on maternal and paternal lineage tracing for at least three generations, available lipid profile records, and body mass index (BMI).

This study was conducted in accordance with the Helsinki Guidelines and ethical approval was obtained from the Ministry of Health for sample collection (2015/242). Informed consent was obtained from all volunteers.

### 4.2. Total gDNA Extraction and Analysis

Total gDNA was extracted from 5 mL whole blood using a salt extraction method [38]. Isolated DNA was analyzed using Nanodrop spectrophotometry (Thermo Fisher Scientific, Waltham, MA, USA) at 260 and 280 nm for concentration and purity. The extracted gDNA was diluted to give a final concentration of 5 ng/µL at −20 °C for further analysis. 

### 4.3. APOCII Sequencing: Primer Design and Polymerase Chain Reaction (PCR) Amplification

Different primer sets (*n* = 10) were designed to flank a specific target region (Figure 7) throughout the *APOCII* locus (4.980 kbp) based on the complete published human reference sequence in GenBank (http://www.ncbi.nlm.nih.gov, NCBI; accessed on 1 April 2018) using Primer 3 (v. 0.4.0) [39,40]. The criteria for the primers designed included absence of specified mispairing based on the genomic library, use of 1-based sequence positions, primer length not exceeding 20 nucleotides, a GC content of 45%, and absence of occurrence of secondary priming sites (Table 8). The optimum annealing temperatures for the custom designed primer sets were determined to be 53 °C for six primer sets and 63 °C for four primer sets. 

Amplification by PCR was established and optimized for the target sequence regions across the *APOCII* loci. The PCR amplification steps were as follows: initial denaturation at 95 °C for 5 min; 35 cycles of denaturation at 95 °C for 30 s, annealing at 53 °C and 63 °C for 30 s each, and extension at 72 °C for 30 s; and a final extension at 72 °C for 7 min with a hold at 4 °C. The PCR reaction mix was prepared in a total volume of 20 µL consisting of 10 µL Gene Amp Fast PCR Master Mix (2×), 2.5 µL double distilled water, 1.25 µL forward primer (5 µM), 1.25 µL reverse primer (5 µM), and 5 µL diluted DNA (5 ng).

### 4.4. Sanger Sequencing of the APOCII Target Regions

Nucleospin columns, Nucleospin^®^ Gel, and a PCR Clean-up Kit (740609.250; Macherey-Nagel, Dueren, Germany) were used to purify the PCR products and remove remnants and artifacts. The purified products were sequenced starting at the 3′ end with a forward primer using the BigDye™ Terminator v3.1 Cycle Sequencing Kit (4337455; Applied Biosystems Waltham, MA, USA) for the 10 target regions at the *APOCII* locus along with its upstream and downstream sequences. Denaturation of the generated double-stranded DNA products was achieved using the BigDye XTerminator Purification Kit (4376486; Applied Biosystems). Samples were sequenced using an ABI 3130xl Genetic Analyzer. The sequence data were analyzed using the AB DNA sequencing Analysis Software (version 2.5). For quality assurance, a separate sequencing reaction was performed using a reverse primer on all samples, following the same steps as those for the forward primer. 

All sequences were checked for quality by aligning the sequence data from the products with the forward reactions against those of the reverse primers. Sequences with 95–100% matches between the two compared sequences confirmed the reproducibility and fidelity and were used for sequence analysis and variant identification.

### 4.5. Variant Identification and Characterization

The sequence data from the reverse complement reaction products spanning the full *APOCII* locus and its flanking regions (4980 bp) were compared by multiple sequence alignment against the published reference sequence in the GenBank database (NG_008837.1) using AB SeqscapeA software, version 6 (https://www.fishersci.fi/shop/products/seqscape-software-v3-0-initial-license/15908206) to assemble the sequence, annotate it, and identify sequence variants, including SNPs and INDELs.

Variants from the 200 sequenced samples were identified by comparing the newly created database file containing the sequence data with those published in the NCBI database, which were assigned reference sequence (rs) numbers at the *APOCII* locus. The Ensembl genome browser (https://asia.ensembl.org/index.html, accessed on 1 April 2020) was used to check the functional role, regulatory regions, and consequences of the browser using the Ensembl Regulatory Build tool based on gene build 89 and genome assembly GRCh38 [41]. 

### 4.6. Genetic and Statistical Analysis 

Genotype and allele frequencies for all 52 identified *APOCII* sequence variants were estimated using a simple gene counting method. Genotype and allele frequencies were tested for HWE using GENEPOP Version 4.2 [39] at a significance level of *p* < 0.05. Haplotype analysis and construction were based on the estimation of LD among the 52 *APOCII* variants identified in the 200 samples sequenced using Haploview 4.2. For LD analysis, the cutoff criteria excluded genetic variants with a MAF < 0.05 or deviated from HWE at *p* < 0.001. Principal component analysis was conducted to investigate the genetic relationships between the studied groups using the prcomp function in R version 4.1 [42].

Allele frequencies for all the variants were calculated for the four studied groups. The differences in allele frequencies among the four groups were calculated to identify variants with different frequency distributions. Pearson’s chi-square test was used to determine significant differences in the genotype distribution between the four sample groups. The *p* value was set at *p* ≤ 0.05.

## 5. Conclusions

This study reports a sequence variant analysis of the *APOCII* locus. These findings add to the ongoing research to identify potential variants that are ethnically specific to various apolipoprotein gene loci and associated LPL genes [24,25,26] that can be used in diagnostic panels for dyslipidemia and metabolic disorders. Identifying potentially associated noncoding variants at these loci would allow the investigation of gene–gene interactions in a large sample cohort. Variant rs5120 was identified as a potential marker for T2DM and HTG. In addition, variant rs71338739 was identified as a potential informative marker for groups with Arab ethnicity that can be used for population stratification and DNA forensics. The limitation of this study is mainly attributed to the sample size, which allowed the identification of variants in the coding region to assess their association with T2DM and HTG. However, establishing association of potential variants was not possible due to the limited sample size.

## Figures and Tables

**Figure 1 ijms-24-16293-f001:**
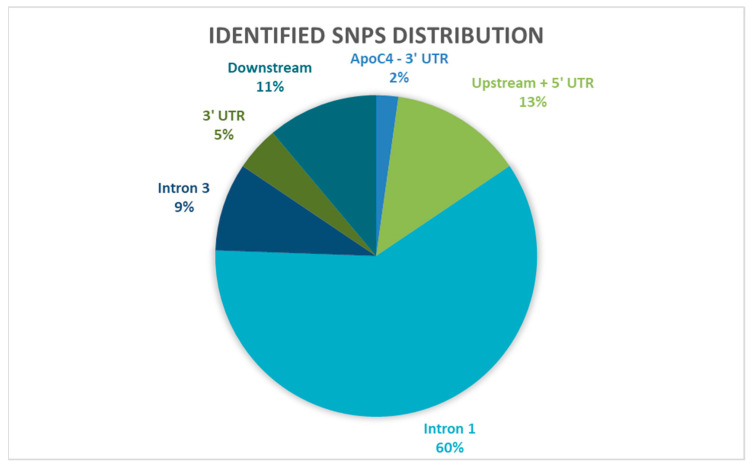
Distribution of identified variants in the different sequenced regions at the *APOCII* locus and its flanking regions.

**Figure 2 ijms-24-16293-f002:**
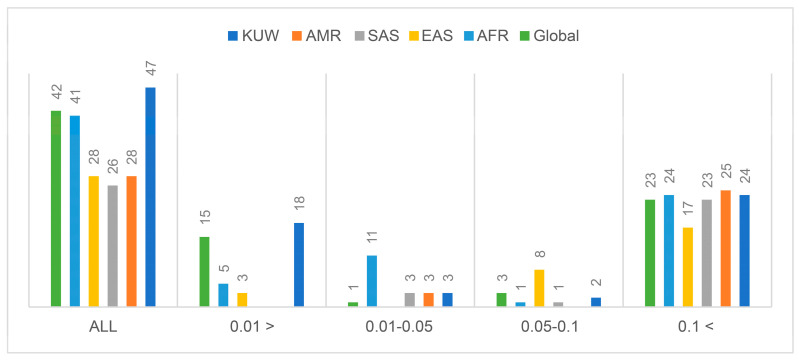
Minor allele frequency (MAF) distribution among the Kuwaiti population compared to that in the global population and four major populations for the identified sequence variants (*n* = 47) excluding the novel variants. The populations illustrated include Kuwait (KUW), admixed Americans (AMR), South Asians (SAS), East Asians (EAS), Africans (AFR).

**Figure 3 ijms-24-16293-f003:**
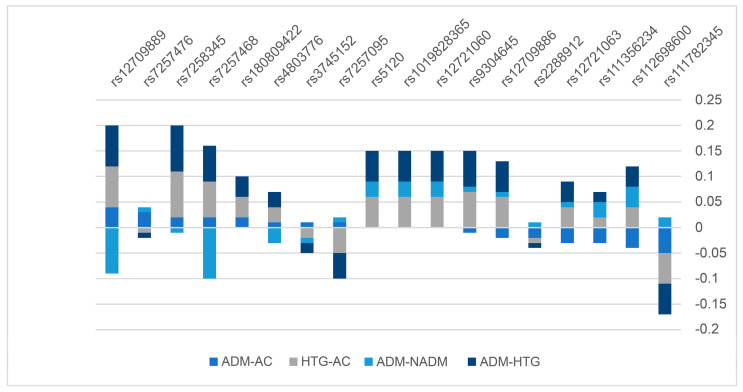
Distribution of the differences in allelic frequencies for 18 of the variants identified by resequencing the full *APOCII* locus in 200 samples as compared between Kuwaiti Arabs with T2DM (ADM) versus controls (AC: blue), HTG versus controls (grey), Kuwaiti Arabs with T2DM versus non-Arabs with T2DM (NADM: light blue) and Kuwaiti Arabs with T2DM versus HTG (dark blue). Distribution of the MAF among the Kuwaiti population is in comparison to that of the global and four major populations for the identified sequence variants (*n* = 47) excluding the novel variants.

**Figure 4 ijms-24-16293-f004:**
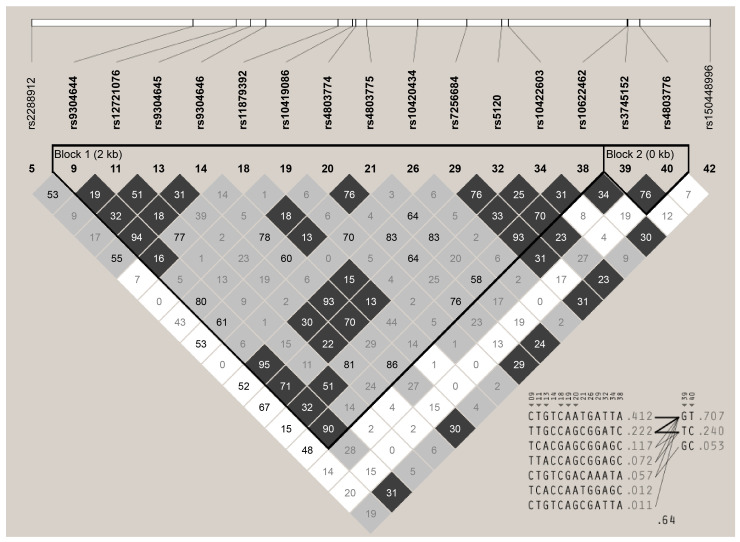
Linkage disequilibrium (LD) plot between 17 common *ApoCII* sequence variants including two insertions. The confidence bounds color scheme shows evidence of LD (dark grey, black), uninformative (shades of light grey), and strong evidence of recombination (white). The r^2^ value is expressed as the percentile inside each pair. The frequency of each haplotype in the block is shown on the right. The SNP numbers across the top of the haplotypes correspond to those in the LD plot and the markers indicate haplotype tagging SNP (htSNP). Only haplotypes with a frequency greater than 1% are shown.

**Figure 5 ijms-24-16293-f005:**
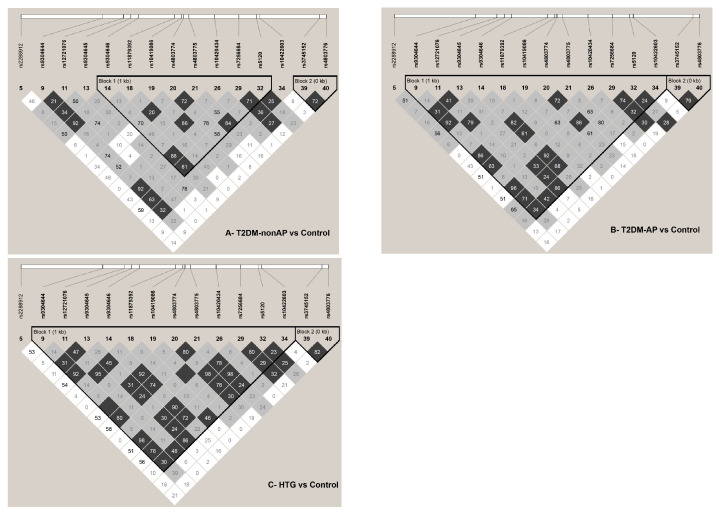
Linkage disequilibrium plot between 15 common *ApoCII* sequence variants. The confidence bounds color scheme shows evidence of LD (dark grey, black), uninformative (shades of light grey), and strong evidence of recombination (white). The r^2^ value is expressed as the percentile inside each pair. The LD plot in (**A**) differs from (**B**,**C**) in the number of informative pairs and representative SNPs constructing the first haplotype block. New SNP pairs show strong LD (r^2^ = 1) defined by the blank dark grey pair in (**C**).

**Figure 6 ijms-24-16293-f006:**
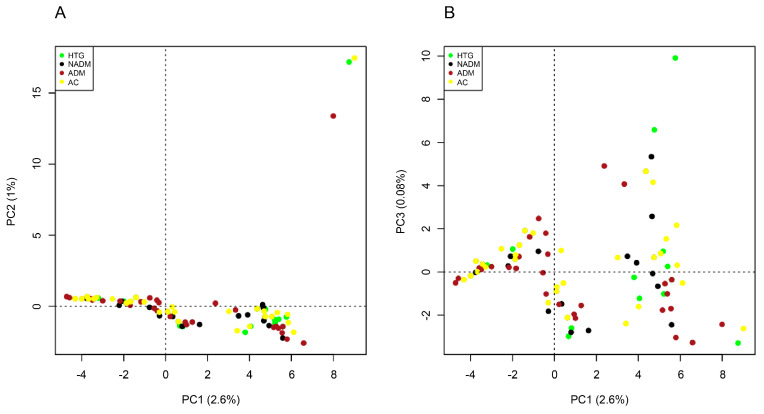
Principal component analysis employing two coordinates: PCA1 (**A**) and PCA2 (**B**). There was no clustering for the studied groups.

**Figure 7 ijms-24-16293-f007:**
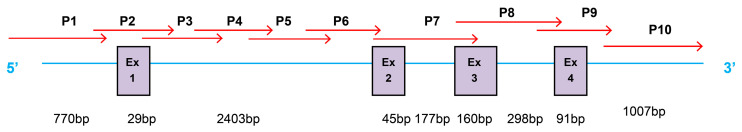
All primers were designed using 10 sets of primers targeting a 413–845 bp sequence at the *APOCII* locus.

**Table 1 ijms-24-16293-t001:** Summary of the 52 identified sequence variants at the full *APOCII* locus and flanking sequences based on genome assembly GRch38.p10.

	SNP ID	Location	Genomic Location	Type	Region
1	rs10425530	g.162 G>A	44945438	SNP	APOC4 3′UTR
2	rs371259300	g.185-184 Del A	44945460	INDEL	APOC4 3′UTR
3	rs111782345	g.496 G>A	44945772	SNP	5′UTR
4	rs112698600	g.538 C>T	44945814	SNP	5′UTR
5	rs111356234	g.584 G>A	44945860	SNP	5′UTR
6	rs12721063	g.633 G>A	44945909	SNP	5′UTR
7	rs2288912	g.666 C>G	44945942	SNP	5′UTR
8	KUA ApoCII 1	g.678 G>A	44945954	SNP	5′UTR
9	rs71338739	g.955_959 Del TGTG	44946192	INDEL	Intron 1
10	rs12709886	g.1160 C>T	44946436	SNP	Intron 1
11	rs9304644	g.1500 C>T	44946776	SNP	Intron 1
12	rs527585098	g.1531 G>A	44946807	SNP	Intron 1
13	rs12721076	g.1719 T>C	44946995	SNP	Intron 1
14	KUA ApoCII 2	g.1776 T>C	44947052	SNP	Intron 1
15	rs9304645	g.1795 G>A	44947071	SNP	Intron 1
16	rs9304646	g.1875 T>C	44947151	SNP	Intron 1
17	rs113329239	g.1878-1879 Del AAA	44947153_44947154	INDEL	Intron 1
18	rs920667500	g.1985 T>C	44947261	SNP	Intron 1
19	rs534276172	g.1986 G>T	44947262	SNP	Intron 1
20	rs11879392	g.2244 C>G	44947520	SNP	Intron 1
21	rs10419086	g.2319 A>G	44947595	SNP	Intron 1
22	rs4803774	g.2339 A>G	44947615	SNP	Intron 1
23	rs4803775	g.2395 T>C	44947671	SNP	Intron 1
24	rs772477913	g.2543 T>C	44947819	SNP	Intron 1
25	rs151176577	g.2569 G>A	44947845	SNP	Intron 1
26	rs12721058	g.2598 C>G	44947874	SNP	Intron 1
27	KUA ApoCII 3	g.2620 A>C	44947896	SNP	Intron 1
28	rs10420434	g.2657 G>A	44947933	SNP	Intron 1
29	KUA ApoCII 4	g.2747 C>T	44948023	SNP	Intron 1
30	rs555315266	g.2802 G>A	44948078	SNP	Intron 1
31	rs7256684	g.2909 A>G	44948185	SNP	Intron 1
32	rs12721060	g.3001 T>G	44948277	SNP	Intron 1
33	rs1019828365	g.3082 C>T	44948358	SNP	Intron 1
34	rs5120	g.3087 T>A	44948363	SNP	Intron 1
35	rs7257095	g.3114 C>G	44948390	SNP	Intron 1
36	rs10422603	g.3123 T>G	44948399	SNP	Intron 1
37	rs569591876	g.3147_3148 Del C	44948423_44948424	INDEL	Intron 1
38	rs373211202	g.3173 C>A	44948449	SNP	Intron 1
39	rs112265403	g.3662 C>T	44948938	SNP	Intron 3
40	rs10622462	g.3736_3737 Ins CCT	44949012_44949013	INDEL	Intron 3
41	rs3745152	g.3738 G>T	44949014	SNP	Intron 3
42	rs4803776	g.3802 T>C	44949078	SNP	Intron 3
43	rs180809422	g.3822 A>C	44949098	SNP	Intron 3
44	rs150448996	g.4162 Ins -/T	44949437_44949438	INDEL	3′UTR
45	KUA ApoCII 5	g.4190 G>C	44949466	SNP	3′UTR
46	rs1130742	g.4279 C>T	44949555	SNP	3′UTR
47	rs10421404	g.4312 G>A	44949588	SNP	downstream
48	rs78403558	g.4325_4239 Del C(T)_4_	44949602_44949604	INDEL	downstream
49	rs7257468	g.4611 C>T	44949887	SNP	downstream
50	rs7258345	g.4618 T>G	44949894	SNP	downstream
51	rs7257476	g.4632 C>T	44949908	SNP	downstream
52	rs12709889	g.4706 G>A	44949982	SNP	downstream

INDEL, insertion deletion; SNP, single nucleotide polymorphism.

**Table 2 ijms-24-16293-t002:** Classification of the sequence variants identified at the *APOCII* locus and flanking sequences.

	Start	End	Size (bp)	Gene Region	Total	INDELs	SNPs	TRS	TRSV
1	1	25	5	*APOCIV* 3′UTR	2	1	1	1	0
2	25	770	745	5′UTR	6	0	6	5	1
3	771	799	29	Exon 1	0	0	0	0	0
4	500	3202	2403	Intron 1	30	3	27	18	9
5	3203	3247	45	Exon 2	0	0	0	0	0
6	3248	3424	177	Intron 2	0	0	0	0	0
7	3425	3584	160	Exon 3	0	0	0	0	0
8	3585	3882	298	Intron 3	5	1	4	2	2
9	3883	3973	91	Exon 4	0	0	0	0	0
10	3974	4980	1007	3′UTR Downstream	3 6	1 1	2 5	1 4	1 1
	1	4980	4980	All	52	7	45	31	14

TRS, transitions; TRSV, transversions.

**Table 3 ijms-24-16293-t003:** List of the identified *APOCII* novel SNVs in the Kuwaiti samples (*n* = 200).

Variant ID	Gene Position	Region	MAF	Sample Group
KUA ApoCII1	g.678 G>A	5′UTR	0.0025	Non-Arab Diabetic
KUA ApoCII2	g.1776 T>C	Intron 1	0.0025	Arab Diabetic
KUA ApoCII3	g.2620 A>C	Intron 1	0.0025	HTG
KUA ApoCII4	g.2747 C>T	Intron 1	0.0025	HTG
KUA ApoCII5	g.4190 G>C	3′UTR	0.0025	Arab Controls

HTG, hypertriglyceridemia.

**Table 4 ijms-24-16293-t004:** Pairs of SNPs with the highest correlation coefficient (r^2^ ≥ 0.80) that indicates their strong LD.

SNP1	SNP2	r^2^ > 0.80	Distance in bp
rs9304644	rs7256684	0.95	1409
rs9304644	rs9304646	0.94	375
rs9304646	rs7256684	0.93	1034
rs7256684	rs10622462 Ins.	0.93	827
rs9304644	rs10622462 Ins.	0.90	2236
rs9304646	rs10622462 Ins.	0.86	1861
rs4803774	rs7256684	0.83	570
rs4803775	rs5120	0.83	692
rs9304645	rs10422603	0.81	1328
rs9304644	rs4803774	0.81	839

**Table 5 ijms-24-16293-t005:** Analysis of *APOCII* haplotype frequency and association in T2DM-non-Arab (*n* = 50) vs. the control (*n* = 50).

Block	Haplotype	Frequency (*n* = 100)	T2DM-nonAP, Control Frequencies	*p* Value
Block 1	TCAATGAT	0.424	0.409, 0.439	0.6682
	CCAGCGGA	0.279	0.200, 0.358	0.0125
	CGAGCGGA	0.14	0.200, 0.080	0.0145
	TCGACAAA	0.075	0.070, 0.080	0.7873
	CCAATGGA	0.021	0.040, 0.002	0.0592
	TCAGCGAT	0.015	0.030, 0.000	0.0864
	TCAGCGGA	0.011	0.000, 0.022	0.1428
	CCAATGAT	0.01	0.020, 0.000	0.1573
Block 2	GT	0.72	0.740, 0.700	0.5287
	TC	0.22	0.180, 0.260	0.1721
	GC	0.06	0.080, 0.040	0.2337

**Table 6 ijms-24-16293-t006:** Reported *APOCII* variants and their association with different metabolic disorders in various populations.

Variant ID	Region	Global MAF	SNP Change	Population MAF	Predicted Consequence	Reference
rs5120	Intron 1	0.67	T>A	African Black 0.185	TC	[10]
rs10422888	3′flanking	0.132	A>G	African Black 0.078	LDL-C & TC	[10]
rs12709885	C4-3′/C2-5′	<0.01	A>T	African Black 0.018	Apo B, apoA1	[10]
rs2288912	C4-3′/C2-5′	T = 0.66	C>A, G, T	African Black 0.258	All LDL related traits	[10]
rs75463753	C2-Intron1	A = 0.02	G>A	African Black 0.108	LDL-C & TC	[10]
APOC2 p. Val40Ter	118del	NA		European <0.01	Severe HTG	[4,17,29]
rs111628497		NA	G>A, C	European <0.01	HTG	[17]
rs5127	3′UTR	0.000	G>T	European <0.01	High HDL-C & apoA1, lower risk of CHD	[8]

CHD, coronary heart disease; NA, not applicable; LDL-C, low-density lipoprotein cholesterol; TC, total cholesterol.

**Table 7 ijms-24-16293-t007:** Clinical and epidemiological features of the study population (*n* = 200).

Sex	Males	Females	Total
Number	94	106	200
Age (years) (Mean ± Std)	46.68 ± 14.5	48.10 ± 12.93	47.43 ± 13.44
BMI *n* (%)			
˂25	24 (25.53)	17 (16.04)	41 (20.5)
25–30	39 (41.49)	39 (36.79)	78 (39.0)
>30	31 (32.98)	50 (47.17)	81 (40.5)

BMI, body mass index.

**Table 8 ijms-24-16293-t008:** Summary of the primers designed and selected for the amplification of target sequences at the *APOCII* locus.

	Primer Name	Forward Primer	Reverse Primer	Size (bp)	AT (°C)
1	*APOCII5′UTR*	TTGTCTGTGGGGACAAGGAC	ACGGGCACAGAGAGGATTTA	766	63
2	*APOCII*F1	GGGGTTGTGGCTGTGGAG	GCTGGGATTACAGGCATGAG	592	53
3	*APOCII*F2	CCACCACACTCCACAAATCA	ACCGCAACCTCTGTCTTCC	746	63
4	*APOCII*F3	CGGATCACTTGATGTCAGGA	CACTGAGCCCAGCCTAGAAG	601	53
5	*APOCII*F4	GGGCAGAGCCCTAAGGTAAC	AGCTGGAATCACAAGCACCT	602	53
6	*APOCII*F5	AGGCAGGCAAATCACTTGAG	TCCAACATTCTGTGATTCTACTCC	596	63
7	*APOCII*F6	AGGCAGGAGAATTGCTTGAA	GGCTAGGCATCTCATCTTGC	608	53
8	*APOCII*F7	CCAGCAAGATGAGATGCCTA	GGGAGCTCAGTCTGAACCTG	602	53
9	*APOCII*F8	CCCTCCTCCCTCTAACCATC	CACTGCTTTATTCCCATGGAC	413	53
10	*APOCII3′UTR*	GGCACTGCTTTTCTGAGGAC	CTCCGGTAGGCCATAAATGA	845	63

AT, annealing temperature.

## Data Availability

The data presented in this study are openly available in GENBANK for the deposited sequence, and the identified SNP variants are *APOCII* [repository numbers MT073872 and BankIt 2306226]. The genotype and allele frequencies for each variant and the minor allele frequency in the study and other major populations are shown in Appendix A.

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
