# Peer review of "Sequence Variant Analysis of the *APOCII* Locus among an Arab Cohort"

_ijms, 2023, doi:10.3390/ijms242216293_

Round 1

Reviewer 1 Report

Comments and Suggestions for Authors

This is a useful communication in the context that we desperately need more information about genomic variants in many populations such as this one.  The basic information in this communication is sound and is presented appropriately.  The authors are, however, hampered by a relatively small sample size (which they acknowledge) and some of the extensions of limited data into domains such as principal components analysis and haplotype analysis may or may not hold up when larger samples are investigated.  Although the authors mention sample size as a weakness, they do not identify those areas that are of limited value as a consequence.

In general, I found both the introduction and the discussion to be longer than necessary and somewhat repetitive.  The paper would benefit from a consideration of shortening both with a goal of focusing on the data rather than the secondary analyses that fail to meet current sample size considerations.

Although this is an issue to be decided by the editor, I wonder if it would not be more appropriate to use the more accurate term SNV, particularly as it relates to variants  below the 0.05 threshold.

A few minor issues need review:  The terms Kuwaiti Arab and Kuwait Arab seem to be used interchangeably; perhaps one or the other could be used throughout.  The term "distribution" of variants (line 125) should perhaps be "frequency" or else how it is used should be clarified.

Figure 2.  Though the population abbreviations are intelligible to readers experienced in genomics, an addition to the legend would be friendly.  [I note that they are identified in the discussion, but that is a bit late.]

line 183:  "according to" and "based upon" are redundant

Comments on the Quality of English Language

no comments

Author Response

We thank the reviewer for their time, effort and positive feedback that are highly appreciated. The issues pointed out had been corrected in the revised manuscript (highlighted by track review) and are as follows:

  1. The authors are, however, hampered by a relatively small sample size (which they acknowledge) and some of the extensions of limited data into domains such as principal components analysis and haplotype analysis may or may not hold up when larger samples are investigated.  Although the authors mention sample size as a weakness, they do not identify those areas that are of limited value as a consequence.

Author's Response:  We thank the reviewer for their valuable time, effort and comments. In the original manuscript it was mentioned one consequence on lines 279-281 and on line 322-324. Nonethless, we have added the consequence of the limited sample size in the conclusion (highlighted by trackreview); Lines 162-463.

  1. In general, I found both the introduction and the discussion to be longer than necessary and somewhat repetitive.  The paper would benefit from a consideration of shortening both with a goal of focusing on the data rather than the secondary analyses that fail to meet current sample size considerations.

Author's Response:  We have carefully revised the introduction and deleted some statements on lines 47, 58-60, 62-63, 70-71, and 76-77.  We also revised the discussion for redundancy and deleted lines 260-261, 265-266, Lines 314-317 and moved lines 317-318 to 398-3-299, .

  1. Although this is an issue to be decided by the editor, I wonder if it would not be more appropriate to use the more accurate term SNV, particularly as it relates to variants below the 0.05 threshold.

Author's Response:  We thank the reviewer for pointing out defining the variants and haver replaced SNP's with SNV for the variants with a frequency of lower than 1%. We introduced the abbreviation "SNV" on line 99 in the revised manuscript. In addition, we replaced "SNPs" with "variants" when referring to all the variants identified regardless of their frequency. The changes are shown in track review in the revised manuscript.

Reviewer's Minor Issues:

  1. The terms Kuwaiti Arab and Kuwait Arab seem to be used interchangeably; perhaps one or the other could be used throughout.

Author's Response:  We thank the reviewer for this oversight and have corrected "Kuwait Arabs" to "Kuwaiti Arabs" on Line 90, Results section.

  1. The term "distribution" of variants (line 125) should perhaps be "frequency" or else how it is used should be clarified.

Author's Response:  Corrected.

  1. Figure 2.  Though the population abbreviations are intelligible to readers experienced in genomics, an addition to the legend would be friendly.  [I note that they are identified in the discussion, but that is a bit late.]

Author's Response:  Added a statement (Lines 141-142) to define the population's abbreviations.

  1. line 183:  "according to" and "based upon" are redundant

Author's Response:  Corrected and is now line 174.

Reviewer 2 Report

Comments and Suggestions for Authors

The study “Sequence variant analysis of the APOCII locus among an Arab cohort” aimed to identify and characterize the apolipoprotein CII (ApocII) genetic variants by sequencing the complete APOCII locus and its flanking sequences in a sample of the Kuwaiti Arab population, including patients with type 2 diabetes mellitus (T2DM) and hypertriglyceridemia, non-Arab patients with T2DM and healthy Arab controls. The obtained results add to the ongoing research that attempts to identify ethnicity-specific variants in the apolipoprotein gene loci and associated lipoprotein lipase (LPL) gene to elucidate the molecular mechanisms of metabolic disorders.

The entire manuscript is correctly written with a sufficient introductory section, well-explained Materials and Method section, and appropriately presented, well-illustrated, and substantially explained Result section. The obtained data are adequately described in connection with previously published data, and a corresponding and up-to-date reference list accompanies the manuscript.

However, some minor obstacles need to be corrected or explained for the manuscript to be ready for publication.

These include:

Line 360-363… The authors have written: “The inclusion criteria for the full cohort (n = 1000) were documented metabolic disorders at the time of sample collection, known ancestry based on maternal and paternal lineage tracing for at least three generations, available lipid profile records, and body mass index (BMI).” This part is confusing. Please explain the number (n= 1000) of the full cohort.

Line 374-376… The authors have written: “Intact high molecular weight gDNA was diluted to give a final concentration of 5 ng/µL at -20 °C for further analysis.” Please explain how the intact high molecular weight DNA was selected for further analysis or modify the sentence accordingly.

A minor revision of the manuscript is suggested.

Author Response

We thank the reviewer for their time, effort and positive feedback that are highly appreciated. The minor issues pointed out had been corrected in the revised manuscript (highlighted by track review) and are as follows:

  1. Line 360-363… The authors have written: “The inclusion criteria for the full cohort (n = 1000) were documented metabolic disorders at the time of sample collection, known ancestry based on maternal and paternal lineage tracing for at least three generations, available lipid profile records, and body mass index (BMI).” This part is confusing. Please explain the number (n= 1000) of the full cohort.

Author's Response:  This was an oversight of a typing error. It is now corrected to 200.

  1. Line 374-376… The authors have written: “Intact high molecular weight gDNA was diluted to give a final concentration of 5 ng/µL at -20 °C for further analysis.” Please explain how the intact high molecular weight DNA was selected for further analysis or modify the sentence accordingly.

Author's Response:  Amended by replacing "Intact high molecular weight" with "The extracted".